# The Impact of Physical Activity on Depression, Anxiety, and Stress during Pregnancy in Saudi Arabia: A Cross-Sectional Study

**DOI:** 10.3390/medicina60081263

**Published:** 2024-08-05

**Authors:** Abdulaziz M. Alfaqih, Ahmad Y. Alqassim, Mohammed H. Hakami, Ahmed M. Sumayli, Nawaf E. Bakri, Shorog A. Alhazmi, Amal M. Ageeli, Remas A. Kobaice, Nasser A. Hakami, Abdullah Fouad Hamadah, Alanoud M. Masmali, Alhassan H. Hobani

**Affiliations:** 1Saudi Board of Preventive Medicine and Public Health, Ministry of Health, Jazan 45142, Saudi Arabia; 2Family and Community Medicine Department, Faculty of Medicine, Jazan University, Jazan 45142, Saudi Arabia; aalqassim@jazanu.edu.sa; 3Al Madaya Primary Health Care, Ministry of Health, Jazan 45142, Saudi Arabia; mohuhakami@moh.gov.sa; 4College of Medicine, Jazan University, Jazan 45142, Saudi Arabia; ahmedsumayli01@gmail.com (A.M.S.); nawafbakri@hotmail.com (N.E.B.); shorog456@gmail.com (S.A.A.); amalageely0123@gmail.com (A.M.A.); remasali3.03@gmail.com (R.A.K.); anoud62249@gmail.com (A.M.M.); 5Surgery Department, College of Medicine, Jazan University, Jazan 45142, Saudi Arabia; nahakami@jazanu.edu.sa; 6Department of Obstetrics and Gynecology, Maternity and Children Hospital, Makkah 21955, Saudi Arabia; drabdalla.f@gmail.com

**Keywords:** depression, anxiety, stress, physical activity, pregnancy, mental well-being, PPAQ, DASS-21, Saudi Arabia

## Abstract

*Background and Objective*: This study aimed to explore the impact of physical activity on depression, anxiety, and stress among pregnant women in the Jazan region of Saudi Arabia. *Materials and Methods*: A descriptive cross-sectional study was conducted among pregnant women attending randomly selected prenatal clinics in primary healthcare hospitals in Jazan, Saudi Arabia. The calculated sample size was 350. Data were collected conveniently through a semi-structured questionnaire covering demographic details, pregnancy-related characteristics, physical activity assessed using the Pregnancy Physical Activity Questionnaire (PPAQ), and mental health parameters evaluated by the Depression, Anxiety, and Stress Scale-21 (DASS-21). The statistical analyses included descriptive statistics and Wilcoxon and Kruskal–Wallis rank sum tests, with significance levels set at *p* < 0.05. *Results*: The study involved 406 pregnant females. Nearly a third (31%) had a family history of depression, anxiety, or distress. The prevalence of depression, anxiety, and stress was 62.6%, 68.7%, and 38.4%, respectively. The mean sedentary, light, moderate, vigorous, and total energy expenditures were 1.512, 24.35, 22.32, 4.84, and 53.02 metabolic equivalent tasks/day. Anxious females exhibited higher light activity (median 24, *p* = 0.033), while stressed ones showed higher light (median 25, *p* = 0.039), moderate (median 20, *p* < 0.001), and vigorous activity (median 3, *p* < 0.001). A significant association was observed between total energy expenditure and stress levels (*p* < 0.001). *Conclusions*: This study underscores the importance of physical activity in managing depression, anxiety, and stress among pregnant women in Jazan, Saudi Arabia. The findings suggest a need for tailored interventions to promote physical activity to improve mental well-being during pregnancy.

## 1. Introduction

Pregnancy—although a physiological process—is a stressful condition that can be associated with a lot of psychological changes; among the most commonly investigated are anxiety, depression, and stress disorders. These disorders are most commonly reported during the third trimester; yet, when symptoms appear in the first and middle trimesters, they are shown to be associated with more evident disorders in the late trimester [1]. Prenatal psychiatric disorders measured by the Diagnostic and Statistical Manual of Mental Disorders in the United States of America showed a 9.9% prevalence of depressive disorders among pregnant women [2]. The prevalence of anxiety in early pregnancy was estimated to be 15.6% in another study [3]; the occurrence of anxiety symptoms during pregnancy is associated with an increased fear of giving birth and choosing cesarean section instead of vaginal delivery. In Brazil, the prevalence of anxiety disorders among pregnant women reaches up to 26.8%, with a higher prevalence during the third trimester [4]. In Jeddah, Saudi Arabia, a study showed a prevalence of 54%, 37%, and 25% for anxiety, depression, and stress during pregnancy, respectively [5].

Exercise and various forms of physical activity were extensively investigated for decreasing pregnancy-associated psychological changes as effective non-pharmacological interventions. Reviews on this issue demonstrated the positive effect of different exercises (aerobics, yoga, etc.) on improving women’s mental health [6]. Women who are inactive during pregnancy have a 16% higher probability of suffering prenatal depression. Exercise may better reduce postpartum depression than psychosocial interventions [7]. However, randomized controlled trials were initiated during the last decade to investigate this area thoroughly. Intervention (exercise) groups showed more reduction in depressive symptoms compared with control groups. Exercise helps not only in reducing depression, anxiety, and stress but also decreases the incidence of organic physical illnesses, such as gestational diabetes, hypertensive disorders, and fetal macrosomia [8,9,10]. This study aims to determine the association between physical activities and depression, anxiety, and stress among pregnant women in the Jazan region of Saudi Arabia. Our research hypothesis states that inactive pregnant women are more prone to mental and psychiatric illness (depression, anxiety, and stress).

## 2. Materials and Methods

### 2.1. Study Design, Setting, and Population

This study employed a descriptive cross-sectional design, focusing on pregnant women attending prenatal clinics within primary healthcare hospitals in Jazan, Saudi Arabia. Participants were recruited regardless of their stage of pregnancy, provided they consented to participate. Exclusions were made for those who declined participation or did not visit the selected clinics. Jazan has 177 prenatal clinics, each associated with a Primary Health Care Center. From these, we randomly chose eight clinics for inclusion. Typically, these clinics operate two days per week. We included all pregnant women attending these clinics on the specified days who agreed to participate in the study.

### 2.2. Sample Size

The sample size calculation utilized the Raosoft calculator (Raosoft Inc., Seattle, WA, USA, www.raosoft.com, accessed on 14 December 2023), considering a 95% confidence level, a 5% margin of error, and an estimated 20% nonresponse rate. The minimum sample size was determined to be 350 pregnant women. However, we opted for a slightly larger sample of 406 respondents to enhance the reliability of the results.

### 2.3. Data Collection Tool and Procedure

Data collection was conducted during January and February 2023, utilizing Google Forms for data input. An interview-based semi-structured questionnaire was administered, with the data collector responsible for completing the questionnaire during the interview process. It contained four sections, with the first assessing demographic details, such as age, nationality, residency, education, occupation, and income. The second it assessed pregnancy-related characteristics, including body mass index, smoking habit, history of depression/anxiety/distress, hypertension, diabetes mellitus, family history of depression/anxiety/distress, current week of pregnancy, and history of mother-related or fetus-related complications in previous pregnancies.

The third section of the questionnaire assessed physical activity using the Pregnancy Physical Activity Questionnaire (PPAQ), initially developed by Chasan-Taber et al. [11]. The PPAQ comprises 33 different activities categorized into 16 household and caregiving activities, three transportation- and inactivity-related activities, nine sports-related activities, and five occupational activities. Participants indicated the closest amount of time they spent on these activities daily or weekly over the preceding three months, choosing from six options: never, less than half an hour per day, between half and one hour per day, one and two hours per day, two and three hours per day, or more than three hours per day. Additionally, two open-ended questions let participants specify sports activities not listed and practiced weekly. Papazian T et al. validated the Arabic version of PPAQ [12].

The fourth section employed the Depression, Anxiety, and Stress Scale-21 (DASS-21), developed by Lovibond and Lovibond [13]. This scale features three subscales, each with seven questions rated on a four-point Likert scale ranging from “Strongly Disagree” (0) to “Totally Agree” (3). The Arabic version of the DASS-21 was validated [14]. It showed a commendable reliability coefficient of 0.76 for depression, 0.75 for anxiety, and 0.77 for stress.

### 2.4. Data Processing and Analysis

The data were cleaned in an Excel sheet and entered into SPSS software version 27 (Statistical Product and Service Solutions, SPSS Inc., Chicago, IL, USA) for analysis. Descriptive statistics were used to summarize the categorical and continuous variables, including frequencies, percentages, means, and standard deviations. Participants were classified into different metabolic equivalent task (MET) categories: sedentary (<1.5 METs), light (1.5 < 3.0 METs), moderate (3.0–6.0 METs), or vigorous (>6.0 METs). We followed the calculation guidelines described by Chasan-Taber et al. [11], multiplying each self-reported answer of each category by its MET value, as specified by the original author, to reach the total daily energy expenditure (METs × hours/day). Wilcoxon rank sum and Kruskal–Wallis rank sum tests were used to assess the association between energy expenditure during pregnancy with demographic characteristics, pregnancy-related characteristics, and mental health parameters. In contrast, only the Wilcoxon rank-sum test was used to evaluate the association between total energy expenditure during pregnancy and depression, anxiety, and stress. All analyses were two-tailed, with the alpha set at the 0.05 level of significance.

### 2.5. Ethical Consideration

Full approval was obtained from the Jazan Health Ethics Committee, with Reference No: 23140, and a decision date of 28 December 2023. Written informed consent was obtained from participants, and they only participated once they gave full approval. The participants’ privacy and confidentiality were maintained, and their information was used for scientific purposes only.

## 3. Results

The study involved 406 females, with a median age of 28 (interquartile range 24–35). The majority were obese (97%), Saudi (85%), lived in cities (75%), had a university education (62%), were not working (61%), and 50% earned less than 5000 SAR.

Table 1 shows that the majority did not smoke (91%) and had no history of hypertension (84%), diabetes (85%), or depression/anxiety (83%). However, 31% had a family history of depression/anxiety/distress. Regarding pregnancy, participants were at various stages, with a median of 17 weeks (interquartile range 8–28 weeks). Approximately 44% were experiencing their first pregnancy, while 56% had previous pregnancies. Of those with prior pregnancies, 23% had experienced complications, and 31% had encountered fetal-related issues.

Table 2 describes the physical activity levels and energy expenditure among the pregnant females. Sedentary activities ranged from 0 to 3, with a mean of 1.512 and a standard deviation of 1.02. Light physical activities ranged from 0 to 73.5, with a mean of 24.35 and a standard deviation of 10.7. Moderate activities spanned from 0 to 147.9, with a mean of 22.32 and a standard deviation of 21.08. Vigorous activities ranged from 0 to 40.5, with a mean of 4.84 and a standard deviation of 7.34. However, household activities ranged from 0 to 92.7, with a mean of 28.17 and a standard deviation of 14.09. Transportation-related activities ranged from 0 to 24, with a mean of 5.11 and a standard deviation of 4.39. Occupational activities ranged from 0 to 42.3, with a mean of 5.24 and a standard deviation of 8.31. Sports-related activities ranged from 0 to 105.9, with a mean of 14.59 and a standard deviation of 17.55. Finally, the total energy expenditure ranged from 0 to 264.9, with a mean of 53.02 and a standard deviation of 34.27.

Table 3 describes the predictors of total energy expenditure during pregnancy. Postgraduate females displayed more moderate (median 39, *p*-value < 0.091) and vigorous activity (median 8, *p*-value = 0.009) than other education groups. Moreover, working females demonstrated higher levels of moderate (median 18, *p*-value = 0.036) and light activity (median 25, *p*-value = 0.001) compared with their nonworking counterparts. Smokers showed more sedentary activity (median 2.50, *p* < 0.001).

Females with hypertension engaged in more light (median 27, *p* = 0.001) and moderate activity (median 21, *p* = 0.001). Similarly, diabetic females exhibited more moderate activity (median 22, *p* = 0.004). Females with a family history of depression/anxiety/distress engaged in less vigorous behavior (median 0, *p* < 0.001).

Females in their first pregnancy reported less light activity (median 21, *p* = 0.006) than those with previous pregnancies (median 18). Females experiencing fetus-related complications engaged in less moderate activity (median 11, *p* = 0.005). Anxious females exhibited more light activity (median 24, *p* = 0.033), while stressed ones showed more light (median 25, *p* = 0.039), moderate (median 20, *p* < 0.001), and vigorous activity (median 3, *p* < 0.001).

Figure 1 shows that the prevalence of depression, anxiety, and stress among pregnant women was 62.6%, 68.7%, and 38.4%, respectively. Roughly 12.3% had mild depression, 24.1% had moderate depression, 9.9% had severe depression, and 16.3% had extremely severe depression. Regarding anxiety, 4.7% had mild anxiety, 24.6% had moderate anxiety, 11.3% had severe anxiety, and 28.1% had extremely severe anxiety. Regarding stress, 10.1% had mild stress, 9.4% had moderate stress, 12.6% had severe stress, and 6.4% had extremely severe stress. Figure 2 illustrates that total energy expenditure was significantly associated with stress, with stressed females having higher energy expenditure (mean = 60.26 vs. mean = 48.5).

## 4. Discussion

This study aimed to investigate the effect of physical activity during pregnancy on depression, anxiety, and stress among Saudi women. Regarding types of physical activity and energy expenditure among pregnant women, the mean for sedentary activities was 1.5, the mean for light physical activity was 24.3, moderate activity was 22.3, and the mean for vigorous activity was 4.8. Furthermore, the mean for household activity was 28.1. For transportation-related activity, the mean was 5.1, and the mean of sports-related activity was 14.5. The mean total energy expenditure was 53. In comparison, a study conducted among Polish women reported significantly higher average hours of physical activity: 94.98 h for light activity, 55.56 h for moderate activity, and 0.97 h for vigorous activity [15]. The disparities between these findings may be attributed to differences in cultural attitudes towards physical activity, varying standards of activity levels, and differences in the study populations.

The study showed that energy expenditure changes greatly during pregnancy, where the basal metabolic rate increases at a rate of 10.7 kcal/gestational week, and the total energy expenditure increases by 5.2 kcal/gestational week. Guidelines recommend that women increase their physical activity during pregnancy to more than 16 MET hours per week; this can be achieved by walking at 3.2 km per hour for about 6.4 h per week or exercising on a stationary bicycle for about 2.7 h per week [15,16,17,18].

A controlled trial suggested integrating a supervised group exercise during prenatal care programs. This was shown to increase psychological well-being and decrease the incidence of depression in women at risk during pregnancy and during the postpartum period [19].

In this study, 36.4% had mild to moderate depression, and 26.2% had severe to extremely severe depressive symptoms. Moreover, 4.7% had mild anxiety, 24.6% showed moderate anxiety, and 39.4% turned out to have severe to extremely severe anxiety. Regarding stress, 10.1% had mild stress, 9.4% had moderate stress, and 19% showed severe or extremely severe stress. Compared with the epidemiology shown in the literature, the prevalence of depression during pregnancy was estimated to be around 9.9% in the United States [2]. There was a 15.6% prevalence of anxiety and 25% prevalence of stress in Jeddah, Saudi Arabia [5].

Different exercise modalities are suggested to decrease these symptoms both during pregnancy and after giving birth [20,21]. A review article concluded that there was a significant reduction in postpartum depression and anxiety among women who exercised during pregnancy [22]. Another review article suggested that exercise of about 644 MET-min/week is needed to moderately affect the incidence and severity of prenatal depression [23]. Exercise during pregnancy is not only beneficial for pregnant mothers, but there is also evidence of infant benefits. These benefits include but are not limited to lower birth weight, decreased risk of prematurity, and better newborn neurodevelopmental aspects [24]. These results—which align with the literature—directly impact women’s and newborns’ health.

This study is limited by its lack of control groups to accurately assess the effect size of exercise in decreasing depression, anxiety, and stress during pregnancy. It is also limited due to not controlling for confounders such as baseline Body Mass Index (BMI) and not considering the high prevalence of obesity in Saudi Arabia and the Middle East. These factors may have directly impacted the outcome variables and thus affected the interpretation of the results.

## 5. Conclusions

The study revealed that more than half of our study participants showed some anxiety and depressive symptoms, and more than a third expressed experiencing stress during pregnancy. For instance, participants showed a wide range of energy expenditure related to different physical activities. The mean of sedentary activities was 1.5, the mean of light physical activity was 24.3, and the mean of vigorous physical activity was 4.8. There was strong evidence supporting the positive effect of exercise during pregnancy both on the mental health of the mother and the physical health of the infants. As a result, we encourage women in all pregnancy trimesters to engage in a regular exercise schedule corresponding to their previous fitness level; group classes are also encouraged.

## Figures and Tables

**Figure 1 medicina-60-01263-f001:**
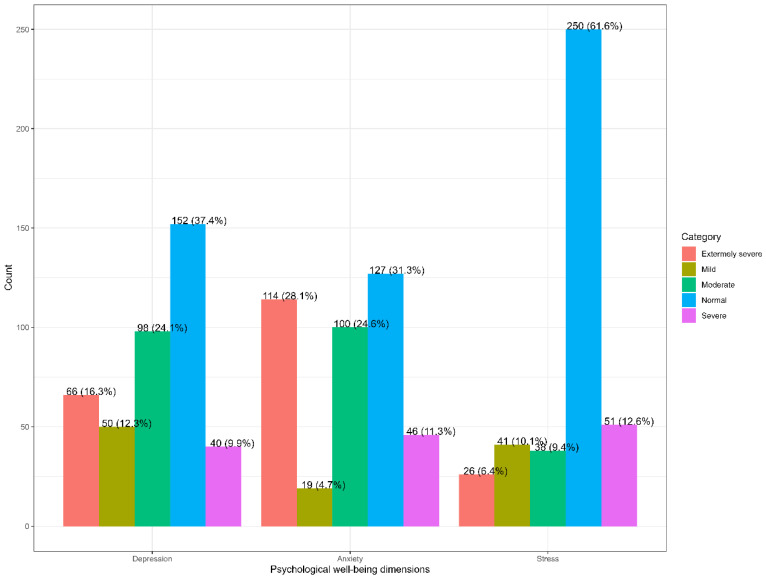
Prevalence of depression, anxiety, and stress during pregnancy.

**Figure 2 medicina-60-01263-f002:**
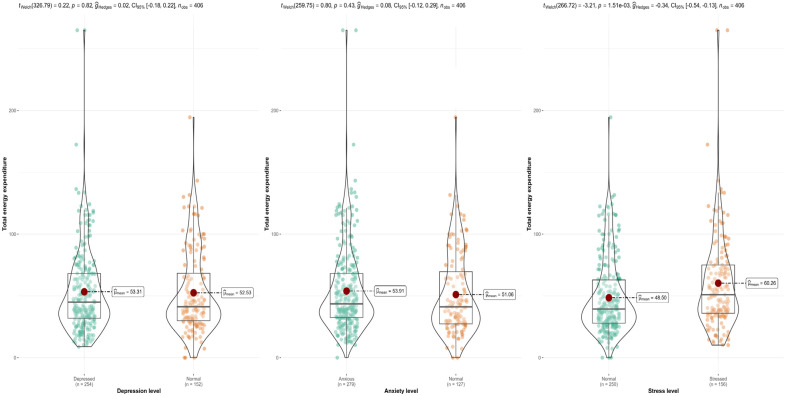
Association between total energy expenditure during pregnancy and depression, anxiety, and stress.

**Table 1 medicina-60-01263-t001:** Sociodemographic and health features of the study cohort.

Characteristic	N = 406 ^1^
**Age in years**	28 (24, 35)
**Body mass index (Kg/m^2^)**	
Obese	395 (97%)
Overweight	11 (2.7%)
**Nationality**	
Non-Saudi	61 (15%)
Saudi	345 (85%)
**Residency**	
Urban	303 (75%)
Rural	103 (25%)
**Level of education**	
Post-graduate	31 (7.6%)
Primary school	19 (4.7%)
Secondary school	103 (25%)
University	253 (62%)
**Work status**	
Not working	249 (61%)
Working	157 (39%)
**Household income**	
Less than 5000 SAR	203 (50%)
5000–9999 SAR	114 (28%)
More than 10,000 SAR	89 (22%)
**Smoking**	
No	368 (91%)
Yes	38 (9.4%)
**Comorbidities**	
**Hypertension**	65 (16%)
**Diabetes mellitus**	60 (15%)
**Depression/anxiety/distress**	71 (17%)
**Family history of depression/anxiety/distress**	
No	282 (69%)
Yes	124 (31%)
**Current week of pregnancy**	17 (8, 28)
**First pregnancy**	
No	226 (56%)
Yes	180 (44%)
**History of mother-related complications in previous pregnancies**	
No	174 (77%)
Yes	52 (23%)
Unknown	180
**History of fetus-related complications in previous pregnancies**	
No	156 (69%)
Yes	70 (31%)
Unknown	180

^1^ Median (Interquartile Range); n (%).

**Table 2 medicina-60-01263-t002:** Distribution of energy expenditure across activities (MET/day).

	Minimum	Maximum	Mean	Standard Deviation	25 Percentile	Median	75 Percentile
**Sedentary**	0.00	3.00	1.512	1.02	0.75	1.50	2.50
**Light**	0.00	73.50	24.35	10.70	17.11	23.65	30.04
**Moderate**	0.00	147.90	22.32	21.08	7.85	15.55	30.33
**Vigorous**	0.00	40.50	4.84	7.34	0.00	3.25	6.75
**Household**	0.00	92.70	28.17	14.09	17.52	25.91	36.65
**Transport**	0.00	24.00	5.11	4.39	2.00	4.00	6.13
**Occupation**	0.00	42.30	5.24	8.31	0.00	0.00	8.4
**Sport**	0.00	105.90	14.59	17.55	2.40	7.95	21.12
**Total energy expenditure**	0.00	264.90	53.02	34.27	30.98	42.41	68.08

**Table 3 medicina-60-01263-t003:** Association between energy expenditure during pregnancy and demographic characteristics, pregnancy-related characteristics, and mental health parameters.

Characteristic	Sedentary Activity		Light Activity		Moderate Activity		Vigorous Activity	
	N = 406 ^1^	*p*-Value ^2^	N = 406 ^1^	*p*-Value ^2^	N = 406 ^1^	*p*-Value ^2^	N = 406 ^1^	*p*-Value ^2^
**Nationality**		0.9		0.5		0.2		0.2
Non-Saudi	1.50 (0.50, 2.50)		24 (18, 30)		15 (9, 21)		0 (0, 5)	
Saudi	1.50 (0.75, 2.50)		23 (17, 30)		16 (8, 35)		3 (0, 8)	
**Residency**		0.2		>0.9		0.6		0.5
City	1.50 (0.50, 2.50)		24 (17, 30)		15 (8, 29)		3 (0, 7)	
Village	1.50 (0.75, 2.50)		25 (19, 28)		17 (8, 38)		3 (0, 9)	
**Level of education**		0.2		0.067		**0.000**		**0.009**
Post-graduate	0.75 (0.75, 1.50)		28 (21, 33)		39 (15, 52)		8 (2, 12)	
Primary school	1.50 (0.25, 2.50)		23 (19, 26)		17 (10, 19)		0 (0, 4)	
Secondary school	1.50 (0.50, 2.50)		23 (15, 28)		13 (8, 22)		3 (0, 5)	
University	1.50 (0.75, 2.50)		24 (17, 30)		16 (7, 33)		0 (0, 7)	
**Work status**		0.4		**0.001**		**0.036**		0.7
Not working	1.50 (0.50, 2.50)		23 (16, 28)		14 (8, 27)		3 (0, 7)	
Working	1.50 (0.75, 2.50)		25 (19, 33)		18 (9, 38)		3 (0, 9)	
**Income**		0.14		0.2		0.12		0.9
5000–9999 SAR	1.50 (0.56, 2.25)		23 (16, 30)		13 (7, 29)		0 (0, 8)	
Less than 5000 SAR	1.50 (0.50, 2.50)		24 (17, 30)		16 (8, 28)		3 (0, 7)	
More than 10,000 SAR	1.50 (0.75, 2.50)		24 (19, 32)		19 (9, 39)		3 (0, 8)	
**Smoking**		**0.000**		0.15		0.2		0.9
No	1.50 (0.50, 2.50)		24 (17, 30)		15 (8, 30)		3 (0, 7)	
Yes	2.50 (1.50, 3.00)		26 (21, 30)		18 (10, 30)		2 (0, 6)	
**History of hypertension**		0.8		**0.001**		**0.001**		0.8
No	1.50 (0.50, 2.50)		23 (17, 29)		14 (8, 29)		3 (0, 7)	
Yes	1.50 (0.75, 2.50)		27 (23, 36)		21 (13, 39)		3 (0, 7)	
**History of diabetes**		0.7		0.054		**0.004**		0.5
No	1.50 (0.75, 2.50)		24 (17, 30)		15 (7, 29)		2 (0, 7)	
Yes	1.50 (0.75, 2.50)		25 (20, 33)		22 (11, 40)		3 (0, 8)	
**History of depression/anxiety/distress**		0.9		0.5		0.8		0.6
No	1.50 (0.50, 2.50)		24 (17, 31)		15 (8, 32)		3 (0, 7)	
Yes	1.50 (0.75, 2.50)		23 (19, 27)		17 (9, 29)		0 (0, 7)	
**Family history of depression/anxiety/distress**		**0.002**		0.6		0.2		**0.000**
No	1.50 (0.50, 2.50)		24 (17, 30)		17 (8, 34)		3 (0, 8)	
Yes	1.50 (0.75, 2.50)		24 (17, 30)		14 (8, 27)		0 (0, 4)	
**Week of Pregnancy**		0.5		0.14		>0.9		0.7
Less than 20 weeks	1.50 (0.75, 2.50)		23 (17, 30)		16 (8, 29)		3 (0, 7)	
≥20 weeks	1.50 (0.75, 2.50)		24 (18, 31)		15 (8, 34)		0 (0, 7)	
**First pregnancy**		0.5		**0.006**		0.3		**0.026**
No	1.50 (0.75, 2.50)		25 (18, 31)		15 (8, 26)		0 (0, 7)	
Yes	1.50 (0.75, 2.50)		21 (15, 29)		17 (8, 34)		3 (0, 9)	
**History of complications in previous pregnancies**		0.2		0.4		0.7		0.2
No	1.50 (0.75, 2.50)		25 (17, 32)		15 (8, 29)		0 (0, 7)	
Yes	1.13 (0.69, 2.50)		25 (22, 30)		16 (9, 22)		0 (0, 6)	
Unknown	180		180		180		180	
**History of fetus-related complications in previous pregnancies**		0.3		0.2		**0.005**		**0.033**
No	1.50 (0.75, 2.50)		25 (18, 33)		16 (9, 34)		0 (0, 8)	
Yes	1.50 (0.50, 2.50)		25 (18, 30)		11 (6, 22)		0 (0, 4)	
Unknown	180		180		180		180	
**Depression**		0.6		0.6		>0.9		0.4
Yes	1.50 (0.50, 2.50)		24 (17, 30)		16 (8, 29)		3 (0, 7)	
No	1.50 (0.75, 2.50)		24 (16, 30)		15 (8, 34)		0 (0, 7)	
**Anxiety**		0.2		**0.033**		0.7		>0.9
Yes	1.50 (0.50, 2.50)		24 (18, 30)		16 (8, 29)		3 (0, 7)	
No	1.50 (0.75, 2.50)		21 (15, 29)		15 (7, 35)		3 (0, 8)	
**Stress**		0.3		**0.039**		**0.000**		**0.000**
Yes	1.50 (0.56, 2.50)		23 (16, 29)		13 (7, 28)		0 (0, 5)	
No	1.50 (0.75, 2.50)		25 (19, 32)		20 (10, 37)		3 (0, 9)	

^1^ Median (IQR). ^2^ Wilcoxon rank sum test; Kruskal–Wallis rank sum test.

## Data Availability

The data from this study are available upon request from the corresponding author.

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
