# Peer review of "The Impact of Physical Activity on Depression, Anxiety, and Stress during Pregnancy in Saudi Arabia: A Cross-Sectional Study"

_medicina, 2024, doi:10.3390/medicina60081263_

Round 1

Reviewer 1 Report

Comments and Suggestions for Authors

Thank you for submitting the article

Comments on the Quality of English Language

Author Response

Thank you for your extensive comments. We have thoroughly addressed them and made comprehensive improvements throughout the manuscript. Additionally, we have significantly improved citations as requested.

Reviewer 2 Report

Comments and Suggestions for Authors

This study offers valuable information, but it also has some areas for improvement:

1. Control Groups: To more accurately assess the effect of exercise in reducing depression, anxiety, and stress during pregnancy, it would be beneficial to include control groups that do not participate in specific physical activities.

2. Confounding Factors Control: Considering factors such as baseline Body Mass Index (BMI) and other medical conditions could provide a more comprehensive understanding of the results.

3. Sample Diversity: Expanding the sample to include greater population diversity (e.g., different ages, socioeconomic levels, and cultural backgrounds) could enhance the generalizability of the findings.

4. Long-Term Follow-Up: Conducting long-term follow-up after childbirth would allow for evaluating the ongoing effects of exercise on maternal and infant mental health.

Overall, implementing these adjustments could strengthen the validity and applicability of the results. 

In future research in this area, it would be worthwhile to consider using the STAID scale to assess anxiety before and after physical activity. This would allow for a more precise understanding of its influence. Nonetheless, publishing these results would contribute to scientific knowledge in this field. 

Author Response

Thank you for your insightful feedback and recommendations. We appreciate the time and effort you put into reviewing our manuscript. Here are our responses to your comments:

We acknowledge the importance of including control groups to more accurately assess the effects of exercise on reducing depression, anxiety, and stress during pregnancy. While our current study design did not incorporate control groups, we have highlighted this limitation in the revised manuscript and discussed it in the limitations section. Our research will indeed aim to include appropriate control groups to address this concern.

We recognize and agree that controlling for confounding factors, enhancing sample diversity, and incorporating long-term follow-up are important recommendations that should be considered in future research.

Reviewer 3 Report

Comments and Suggestions for Authors

Thank you for the study concerning pregnant women’s health. The topic is not novel but it may be important at author’s local. I address some concerned points as below:

-Sample size is not clear. The prevalence of concerned variable is missed.

-The study calculated 350 participants. However, the final data was 406 cases. Please explain.

-A study flowchart with the number of inclusion and exclusion cases is required.

-How the metabolic equivalent task (MET) was calculated in this study?

-Results are not present in good way. The current presentation is too long. Please do not duplicate the data in tables.

-In table 1, superscript “1” should not be placed at N=406, similar to other tables. Unit should be provided for age (years), BMI (kg/m2).

-Income was calculated based on pregnant women or both partners. Please define.

-The statistical tests should be provided as footnote under the table.

-The presence of data in table 2-3 is unclear.

-Figure 2 is not good at quality.

-The first paragraph of discussion should mention the main findings and compared to the literature.

-The conclusion is too long. Please focused.

-English editing is strongly required. For example: The calculated sample size was 350. The study involved 406 females, primarily obese (97%), at a median age of 28. The majority did not smoke (91%), and 31% had a family history of depression/anxiety/distress.

-Reference style is not followed the journal’s instruction.

-The manuscript  did not prepare following manuscript’s template.

Overall, the paper needed to be revised seriously. The study lacks of control group (healthy pregnancy) and did not follow the before and after intervention (physical activity). A lot of bias could not be excluded such as the normal pregnancy or abnormal pregnancy. Gestational diabetes could also impact on the pregnancy’s psychological issues accompanied with diet. Thus, the findings are limited.

Best regards,

Comments on the Quality of English Language

-English quality is strongly required for a editing service. 

Author Response

Thank you for the study concerning pregnant women’s health. The topic is not novel but it may be important at author’s local. I address some concerned points as below:

-Sample size is not clear. The prevalence of

concerned variable is missed.

-The study calculated 350 participants. However, the

final data was 406 cases. Please explain.

350 was the minimum acceptable sample size but we opted for a slightly larger sample of 406 respondents to enhance the reliability of the results

-A study flowchart with the number of inclusion and

exclusion cases is required.

We understand that a flowchart illustrating the number of inclusion and exclusion cases would provide clarity on participant selection and the progression of the study. However, due to the nature of our study, we faced certain constraints that prevented us from including a detailed flowchart.

We have, however, provided a detailed description of the participant recruitment and selection process in the revised manuscript to ensure transparency. We appreciate your understanding and hope that this explanation helps clarify why a flowchart was not feasible in this instance.

-How the metabolic equivalent task (MET) was

calculated in this study?

It has been elaborated in the Data Collection Tool and Procedure under methodology hope you find it satisfactory.

-Results are not present in good way. The current presentation is too long. Please do not duplicate the data in tables.

Thank you for your valid concern: To clarify, the detailed presentation in the tables was intended to provide a comprehensive view of the results across various activity levels and characteristics. However, we recognize that this format may have led to an impression of duplication.

In table 3 to be specific we aimed to present detailed data across different activity levels (sedentary, light, moderate, vigorous) for each characteristic. This approach was chosen to allow readers to easily compare the results across these activity levels, rather than repeating the same data in separate sections.

-In table 1, superscript “1” should not be placed at N=406, similar to other tables.

In Table 1, the superscript “1” next to the sample size (N=406) was used to refer to endnotes that provide details about the statistical tests employed in the analysis.

Unit should be provided for age (years), BMI (kg/m2).

This has been modified in the tables

-Income was calculated based on pregnant women or both partners. Please define.

Thank you this involved household income and not that individual respondents. This has been defined

-The statistical tests should be provided as footnote under the table.

 They have been provided in the last row of each table

-The presence of data in table 2-3 is unclear.

The table 2 designed to present the distribution of various characteristics across different levels of physical activity (sedentary, light, moderate, and vigorous) while table 3 compares the characteristics of the pregnant women by their level of physical activity

Round 2

Reviewer 3 Report

Comments and Suggestions for Authors

Dear authors,

Thank you for addressing my concerns.

The paper is improved, however, the manuscript must be revised again. Some points are still not good.

-English editing is strongly required. For example: This sentence is a bad presenatation. "The study involved 406 pregnant females, with a median age of 28 most of whom (97%) were obese."

-Reference citation should be placed following journal's instruction. For example:  "The Arabic version of the DASS-21 was validated (14) and showed a commendable reliability coefficient of 0.76 for depression, 0.75 for anxiety, and 0.77 for stress."

-In table 3, each responses of depression, anxiety, and stress should be the value of YES/NO or Present/absent instead of depressed/anxious/stressed and normal.

-History of diabetes in Table 3 is very common. The readers could not know that is GDM, type I, or II.

Comments on the Quality of English Language

Quality of English is not good throughout the overall paper. Scientific language is strongly needed. For example. Please read again these sentences:

- In Jeddah, Saudi Arabia, a study showed a 54% prevalence of anxiety, a 37% prevalence of depression, and a 25% prevalence of stress during pregnancy (5).

-In comparison, a study conducted among Polish women reported significantly higher averages: light physical activity was 94.98 moderate activity was 55.56, and vigorous activity was 0.97 hours (15).

Author Response

Thank you for your comment. The questionnaire used in this study did not specify the type of diabetes (GDM, type I, or type II) during data collection as we intended to capture a broad picture of diabetes history among the participants without imposing additional categorizations. We acknowledge this limitation and have chosen to present the data as collected without further classification.